# Environmental selection overturns the decay relationship of soil prokaryotic community over geographic distance across grassland biotas

Biao Zhang[1], Kai Xue[1,2,3]*, Shutong Zhou[1], Kui Wang[4], Wenjing Liu[1], Cong Xu[5], Lizhen Cui[4], Linfeng Li[1,6], Qinwei Ran[4], Zongsong Wang[4], Ronghai Hu[1,2], Yanbin Hao[2,4], Xiaoyong Cui[2,4], Yanfen Wang[1,2,7]

[1]College of Resources and Environment, University of Chinese Academy of Sciences, Beijing, China; [2]Yanshan Earth Critical Zone National Research Station, University of Chinese Academy of Sciences, Beijing, China; [3]Key Laboratory of Adaptation and Evolution of Plateau Biota, Chinese Academy of Sciences, Xining, China; [4]College of Life Sciences, University of Chinese Academy of Sciences, Beijing, China; [5]State Key Laboratory of Remote Sensing Science, Aerospace Information Research Institute, Chinese Academy of Sciences, Beijing, China; [6]Environmental Futures Research Institute, School of Environment and Science, Griffith University, Nathan, Australia; [7]State Key Laboratory of Tibetan Plateau Earth System Science (LATPES), Beijing, China

*For correspondence:
xuekai@ucas.ac.cn

Competing interest: The authors declare that no competing interests exist.

**Abstract** Though being fundamental to global diversity distribution, little is known about the geographic pattern of soil microorganisms across different biotas on a large scale. Here, we investigated soil prokaryotic communities from Chinese northern grasslands on a scale up to 4000 km in both alpine and temperate biotas. Prokaryotic similarities increased over geographic distance after tipping points of 1760–1920 km, generating a significant U-shape pattern. Such pattern was likely due to decreased disparities in environmental heterogeneity over geographic distance when across biotas, supported by three lines of evidences: (1) prokaryotic similarities still decreased with the environmental distance, (2) environmental selection dominated prokaryotic assembly, and (3) short-term environmental heterogeneity followed the U-shape pattern spatially, especially attributed to dissolved nutrients. In sum, these results demonstrate that environmental selection overwhelmed the geographic 'distance' effect when across biotas, overturning the previously well-accepted geographic pattern for microbes on a large scale.

## Editor's evaluation

With a massive and extensive survey of soils and prokaryotic communities, this study by Zhang et al. explores the distance-decay relationship in grassland biotas. Their results show that in contrast to the traditional linear relationship within the same biome, a significant U-shape pattern was found across two grassland types. Even more intriguingly, further analysis showed that a homogeneous selection process instead of geographic distance effects may contribute to the soil prokaryotic community patterns. This study will inspire biogeographers, and ecosystem and community ecologists interested in macroecological rules.

## Introduction

To clarify the spatial pattern of biodiversity is one of primary aims in ecology and biogeography (*Bahram et al., 2013*; *Hanson et al., 2012*). In past decades, intensive biogeographic studies have been conducted for macro-organisms, including plants (*Condit et al., 2002*; *Tuomisto et al., 2003*; *Gilbert and Lechowicz, 2004*; *Dexter et al., 2012*; *Zhao et al., 2014*; *Draper et al., 2019*), insects (*Gómez-Rodríguez and Baselga, 2018*; *van der Mescht et al., 2018*), and vertebrates (*Warfe et al., 2013*; *Beisner et al., 2006*). With the emergence and development of next-generation sequencing, increasing attention has been paid recently to the spatial pattern of microorganisms. The similarity of microbial communities has been observed to decrease over geographic distance as that of macro-organisms does, so-called distance-decay relationship, in different habits (e.g., forest *Matsuoka et al., 2016*; *Oono et al., 2017* grassland *Wang et al., 2017a*, desert *Finkel et al., 2012*, and agriculture soils *Jiao et al., 2020*; *Jiao et al., 2019*; *Shi et al., 2018*) for bacteria (*Wang et al., 2017a*; *Zinger et al., 2014*; *Barreto et al., 2014*), archaea (*Jiao et al., 2019*; *Shi et al., 2016*), fungi (*Chen et al., 2019*; *Wang et al., 2017b*; *Yang et al., 2017*; *Zhang et al., 2018*) and specific microbial functional groups (e.g. ammonia-oxidizing archaea, ammonia-oxidizing and sulfate-reducing bacteria *Hu et al., 2015*; *Angermeyer et al., 2016*; *Zhou et al., 2008*; *Zhou et al., 2016*). To certain extent, the reported distance-decay relationship has been regarded as a principal generalization in nature (*Nekola and White, 1999*).

However, no consensus has been reached so far on underlying mechanisms of the distance-decay relationship for soil microbial communities. A few mechanisms for biodiversity maintenance *Zinger et al., 2014* have been proposed to be responsible for such relationship as well, including the environmental heterogeneity, dispersal limitation, and stochastic processes (*Tuomisto et al., 2003*; *Soininen et al., 2007*; *Cottenie, 2005*). Specifically, environmental heterogeneity tends to increase over geographic distance, responsible for the distance-decay relationship (*Hanson et al., 2012*; *Jiao et al., 2019*). Microorganisms have been observed to have the dispersal limitation as macro-organisms do (*Hanson et al., 2012*; *Bell, 2010*; *Lindström and Östman, 2011*), crucial in biodiversity maintenance and evolution (*Cottenie, 2005*; *Leibold et al., 2004*). Spatial configurations and the nature of landscapes influence the dispersal rate of organisms among sites (*Soininen et al., 2007*), and communities tend to be more similar in open and topographically homogeneous settings than in heterogeneous landscapes. Moreover, communities are expected to become increasingly different along with geographic distance as their species are sorted according to their niche requirements (*Bell, 2010*). Under this scenario, dissimilarities among communities parallel to increasing disparities in environmental heterogeneity along with geographic distance. Furthermore, stochastic processes in birth, death, migration, disperse, and drift may also contribute to the distance-decay relationship for soil microbial communities (*Stegen et al., 2012*; *Chase and Myers, 2011*; *Hubbell, 2005*).

The relative importance of various mechanisms is still unclear (*Tuomisto et al., 2003*; *Soininen et al., 2007*; *Cottenie, 2005*), likely being scale- and biota-dependent. Environment heterogeneity has been reported to be more important in influencing the spatial distribution of microorganisms at local scales up to hundreds of kilometers (*Shi et al., 2018*; *Feng et al., 2019*; *Chu et al., 2016*), while dispersal limitation dominated the distance-decay relationship on larger scales (*Green et al., 2004*; *Whitaker et al., 2003*). Moreover, different biotas under distinct climate conditions/latitudes may have different turnover rates for community similarity over geographic distance. High temperature was reported to lead to a lower turnover rate for forest soil microbial community (*Zhou et al., 2016*), likely due to accelerated biochemical reactions and increased ecological niche breadth (*Okie et al., 2015*). Community similarity was observed to decline faster at high than low latitudes on large scales, while the turnover rate was higher at low latitudes on small scales (*Soininen et al., 2007*). However, most previous studies were conducted locally or regionally within a single biota, region or climate type, and surveys for the geographic pattern of soil microorganisms across different biotas are still lacking, which are essential to understand the spatial pattern of microbial communities beyond these scales.

Here, we collected grassland soil samples from two biotas of alpine and temperate grasslands with distinct hydrothermal conditions to investigate the spatial pattern of prokaryotic communities and underlying mechanisms. A total of 258 samples were collected from the top- (0–5 cm in depth) and subsoils (5–20 cm in depth) in both the alpine biota on the Qinghai-Tibet Plateau and the temperate biota on the Inner Mongolia Plateau, China, on a scale up to 4000 km. Our objectives were to test

the following hypotheses: (1) soil prokaryotic community similarity would decrease over geographic distance within and across biotas; (2) the turnover rate of soil prokaryotic community similarity over geographic distance would be higher in the temperate than alpine biota, as temperate biota is with a wider temperature range; (3) the turnover rate of soil prokaryotic community similarity over geographic distance would be lower in top- than subsoil, since the subsoil may be less dynamic as not affected by environmental factors like UV and wind.

## Results

### Prokaryotic and plant community similarity over geographic distance

A total of 11,063 operational taxon units (OTUs) were detected from all 258 grassland soil samples in both alpine and temperate biotas (*Figure 1—figure supplement 1*). A significant (p < 0.001) binomial relationship (U shape) was observed for the prokaryotic community over geographic distance in top- ($R^2 = 0.161$) or subsoil ($R^2 = 0.114$) from all sites on a scale up to 4000 km. Specifically, the prokaryotic community similarity in topsoil decreased over geographic distance on a scale of <1920 km mostly within either temperate or alpine biota, but increased after this tipping point when across biotas (in pairwise sites between alpine and temperate biotas). Similarly, the prokaryotic community similarity in subsoil decreased over geographic distance on a scale of <1760 km mostly within either temperate or alpine biota, but increased after this tipping point when across biotas (*Figure 1*). When across biotas, the prokaryotic community similarity increased significantly over geographic distance with similar slopes (turnover rates) in top- (slope = 0.007, $R^2 = 0.199$) and subsoil (slope = 0.006, $R^2 = 0.134$).

Within the alpine biota, a valid (p < 0.001) distance-decay relationship was observed for the prokaryotic community over geographic distance in top- ($R^2 = 0.034$) or subsoil ($R^2 = 0.013$). However, within the temperate biota, the distance-decay relationship for the prokaryotic community occurred only in topsoil ($R^2 = 0.129$, p < 0.001), while no relationship was observed in subsoil. In topsoil, prokaryotic community similarity had a higher turnover rate in the temperate (−0.012) than alpine biota (−0.005).

Similar to the prokaryotic community, the plant community also exhibited a significant U-shape relationship ($R^2 = 0.071$, p < 0.001) for its similarity over geographic distance in all sites on a scale up to 4000 km, with a tipping point of 1858 km (*Figure 1—figure supplement 2a*). A significant (p < 0.001) distance-decay relationship for plant community was observed within the alpine (*Figure 1—figure supplement 2b*, $R^2 = 0.015$, p < 0.001) or temperate biota (*Figure 1—figure supplement 2b*, $R^2 = 0.005$, p < 0.01).

### Prokaryotic community similarity over environmental distance

Relatively short-term environmental similarity also exhibited a U-shape pattern over geographic distance in either top- or subsoil from all sites on a scale up to 4000 km (*Figure 2a and d*). In contrast, relatively long-term environmental similarity, much higher than short-term environmental similarity on the same scale, did not change greatly over geographic distance in either top- or subsoil.

Soil prokaryotic community similarity decreased significantly (p < 0.001) in all sites over the relatively long-term (turnover rate = −0.291 or −0.278 in top- or subsoil, respectively) or short-term (turnover rate = −0.193 or −0.159 in top- or subsoil, respectively) environmental distance (*Figure 2—figure supplement 1a and e*. In the topsoil, the prokaryotic community similarity decreased significantly (p < 0.001) over the relatively long-term (turnover rate = −0.277, $R^2 = 0.131$) or short-term (turnover rate = −0.194, $R^2 = 0.135$) environmental distance within the alpine biota (*Figure 2—figure supplement 1b*), as well as decreased over long-term (turnover rate = −0.339, $R^2 = 0.108$) or short-term (turnover rate = −0.063, $R^2 = 0.011$) environmental distance within the temperate biota (*Figure 2—figure supplement 1c*). In the subsoil, the prokaryotic community similarity decreased significantly over the relatively long-term (turnover rate = −0.279, $R^2 = 0.104$) or short-term (turnover rate = −0.215, $R^2 = 0.093$) environmental distance within the alpine biota (*Figure 2—figure supplement 1f*), while there was no relationship within the temperate biota (*Figure 2—figure supplement 1g*). Across biotas or in pairwise sites between the alpine cross temperate biota (*Figure 2—figure supplement 1d and h*), the prokaryotic community similarity decreased over relatively long-term (turnover rate = −0.161 or −0.130 in top- or subsoil, respectively) or short-term (turnover rate = −0.191 or −0.175 in top- or subsoil, respectively) environmental distance.

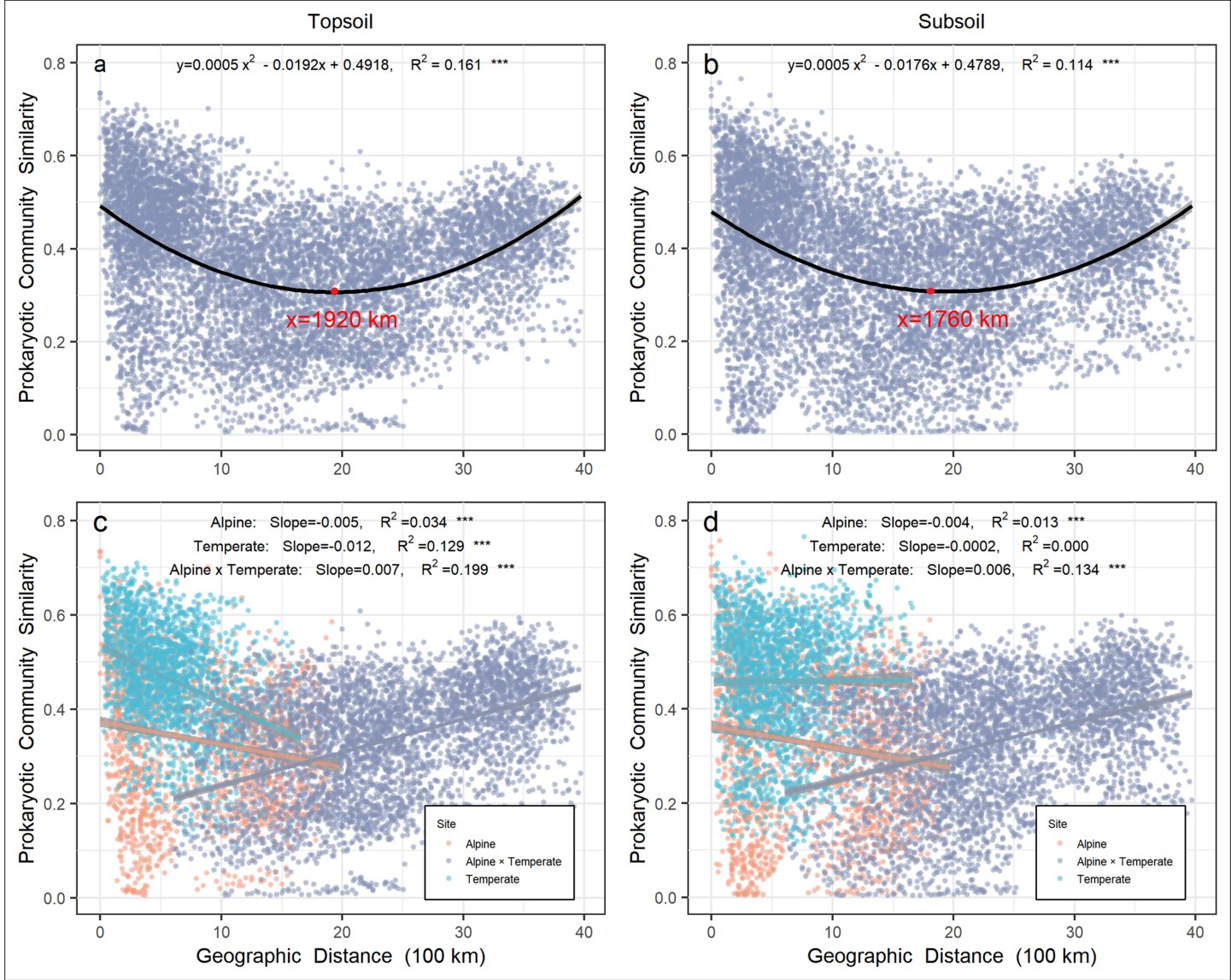

**Figure 1.** Relationship between prokaryotic community similarity over geographic distance in Chinese northern grassland. Panels (**a**) and (**c**) represent the prokaryotic community in topsoil, while panels (**b**) and (**d**) represent the prokaryotic community in subsoil. Orange and light blue points represent pairwise sites within the alpine and temperate biota, respectively. Gray points represent pairwise sites between the alpine biota cross temperate biota. Gray shades stand for 95% confidence interval.

The online version of this article includes the following figure supplement(s) for figure 1:

**Figure supplement 1.** Sampling sites across 1921 km of the alpine grassland in Qinghai-Tibet Plateau (in red) and 1661 km of the temperate grassland in Inner Mongolia (in blue).

**Figure supplement 2.** Relationship between plant community similarity over geographic distance in northern grassland of China.

**Figure supplement 3.** Relationship between prokaryote community over plant community dissimilarity.

**Figure supplement 4.** Rarefaction curve.

By Mantel test, the prokaryotic composition dissimilarity in top- or subsoil was significantly correlated with geographic distance or altitude within the alpine or temperate biota and across the alpine and temperate biotas (*Supplementary file 1*), except that the subsoil prokaryotic composition dissimilarity within the temperate biota was not linked to geographic distance. However, by partial Mantel, the prokaryotic composition dissimilarity in top- or subsoil was not correlated with geographic variables when across biotas (*Table 1*).

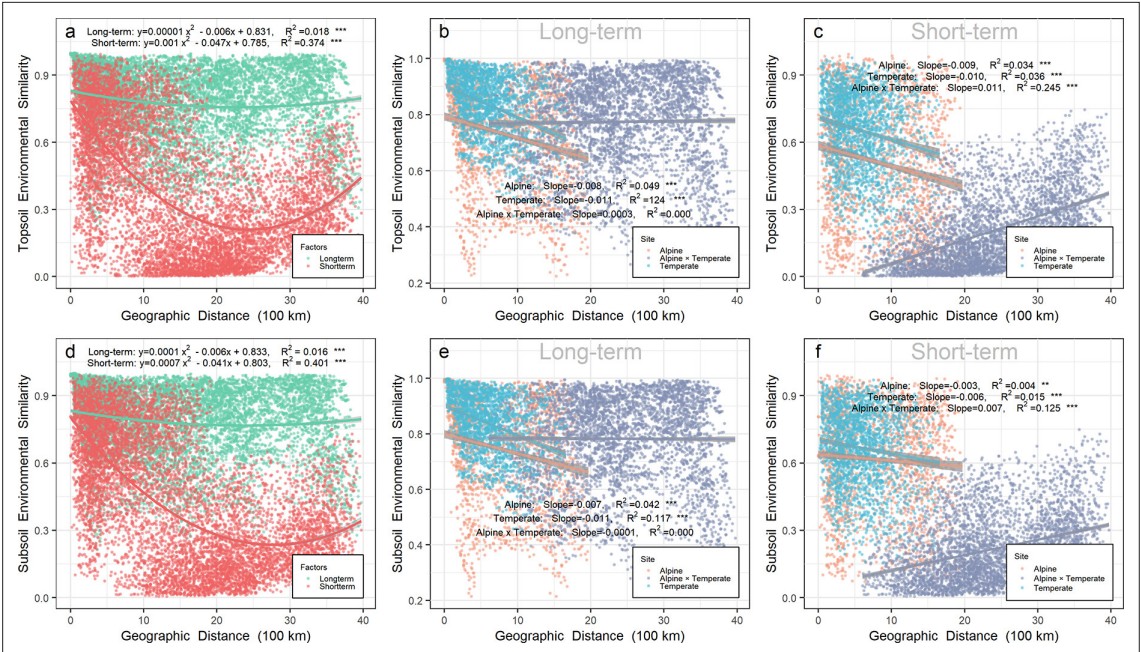

**Figure 2.** Relationship between environmental similarity over geographic distance in Chinese northern grassland. Panels a, b, and c represent the environmental similarity in topsoil, while panels d, e, and f represent the environmental similarity in subsoil. The environmental similarity was calculated by Bray-Curtis index based on relatively long-term (b, e, and green points in a and d) or short-term variables (c, f, and red points in a and d). Relatively long-term environmental variables included mean annual precipitation (MAP), mean annual temperature (MAT), pH, soil organic carbon (SOC), soil total nitrogen (TN) and soil total phosphorus (TP). Relatively short-term environmental variables included soil water content (SWC), soil available phosphorus (AP), dissolved organic carbon (DOC), dissolved organic nitrogen (DON), $NH_4^+$, and $NO_3^-$. Orange, gray, and light blue points represent pairwise sites within the alpine biota, within temperate biota, and alpine biota cross temperate biota, respectively. Gray shades stand for 95% confidence interval.

The online version of this article includes the following figure supplement(s) for figure 2:

**Figure supplement 1.** Distance-decay relationship for prokaryotic community similarity over relative long-term (orange points) and short-term (light blue points) environmental factors.

As revealed by partial Mantel (**Table 1** and **Supplementary file 2**) test, the significant decay relationship between topsoil prokaryotic community similarity and relatively short-term environmental distance across biotas was mainly driven by soil water content (SWC, r = 0.112, p = 0.015), dissolved organic carbon (DOC, r = 0.307, p = 0.001), dissolved organic nitrogen (DON, r = 0.398, p = 0.001) and $NH_4^+$ (r = 0.223, p = 0.001). Similar short-term environmental variables (except $NH_4^+$) were responsible for the significant decay relationship in the subsoil.

Within the alpine biota, the significant distance-decay relationship between topsoil prokaryotic community similarity and relatively long-term environmental distance was driven by mean annual precipitation (MAP, r = 0.235, p = 0.002). Except the relatively long-term variable of MAP (r = 0.199, p = 0.007, **Supplementary file 2**), the short-term environmental variable of DON (r = 0.211, p = 0.029) was also responsible for the significant distance-decay relationship of prokaryotic community similarity in the subsoil within the alpine biota. Within the temperate biota, the significant distance-decay relationship between topsoil prokaryotic community similarity and relatively long-term environmental distance was driven by MAP (r = 0.310, p = 0.001).

## Deterministic and stochastic processes in prokaryotic community assembly

Null model and βNTI (β-nearest taxon index) analyses were used to explore the roles of deterministic and stochastic processes on community assembly. The similarity of observed communities in all samples (**Figure 3—figure supplement 1**) or samples in each biota (**Figure 3—figure supplement 1a and b**) was significantly (p < 0.001) higher than that of the permutated communities generated by the null model, revealing the important role of deterministic processes in determining the community assembly. As shown in **Figure 3**, the range of |βNTI| > 2 also confirmed the importance of

**Table 1.** Partial Mantel test for relationship between prokaryotic community similarity and relatively long-term environmental variables, relatively short-term environmental variables, and geographic variables across biotas.

| | Alpine× temperate | | | |
| | Topsoil | | Subsoil | |
| | r | p | r | p |
|---|---|---|---|---|
| MAP | 0.060 | 0.101 | 0.047 | 0.165 |
| MAT | 0.000 | 0.498 | –0.021 | 0.625 |
| pH | –0.001 | 0.516 | –0.004 | 0.542 |
| SOC | 0.231 | **0.001** | 0.120 | **0.010** |
| TN | 0.151 | **0.003** | 0.046 | 0.146 |
| TP | –0.143 | 0.999 | –0.072 | 0.908 |
| Long-term environment variables | 0.121 | **0.010** | 0.049 | 0.174 |
| SWC | 0.112 | **0.015** | 0.094 | **0.032** |
| AP | 0.003 | 0.387 | 0.064 | 0.124 |
| DOC | 0.307 | **0.001** | 0.206 | **0.001** |
| DON | 0.398 | **0.001** | 0.401 | **0.001** |
| $NH_4^+$ | 0.223 | **0.001** | 0.023 | 0.350 |
| $NO_3^-$ | –0.135 | 0.990 | 0.040 | 0.234 |
| Short-term environment variables | 0.523 | **0.001** | 0.465 | **0.001** |
| Latitude | –0.541 | 1.000 | –0.433 | 1.000 |
| Longitude | –0.525 | 1.000 | –0.430 | 1.000 |
| Distance* | –0.544 | 1.000 | –0.438 | 1.000 |
| Altitude | –0.672 | 1.000 | –0.588 | 1.000 |
| Geographic variables | –0.5664 | 1.000 | –0.462 | 1.000 |

*Calculated by the geographic distance for paired samples based on the longitude and latitude of each sample.

deterministic processes in prokaryotic community assembly, and identified the dominant role (>84%) of homogeneous environmental selection, which means selection under homogeneous abiotic and biotic environmental conditions leading to more similar structures among communities (*Macarthur and We, 1967*). Specifically, the contribution of deterministic processes was relatively lower in the alpine (*Figure 3j*; 92.91% and 87.47% in top- and subsoils, respectively) than temperate (96.63% and 94.04% in top- and subsoils, respectively) biota, and higher in the topsoil than subsoil in all sites. Moreover, most βNTI values were less than –2 either in the top- (86.01–96.63%) or subsoil (84.09–93.37%) from all sites, indicating that prokaryotic communities were assembled mainly by homogeneous selection, in deterministic processes.

We further compared the immigration rates (m) of prokaryotes (*Figure 3—figure supplement 2*) based on the algorithm developed by Hubbell for the neutral theory (*Xing et al., 2019*). Prokaryotic immigration rates were significantly lower in the alpine (0.159 ± 0.008 and 0.146 ± 0.008 in top- and subsoils, respectively, p < 0.01) than temperate biota (0.261 ± 0.010 and 0.246 ± 0.009 in top- and subsoils, respectively, p < 0.01) in the same soil layer. Moreover, immigration rates were higher in the top- (0.159 ± 0.008 and 0.261 ± 0.010 in the alpine and temperate biotas, respectively, p < 0.01) than subsoil (0.146 ± 0.008 and 0.246 ± 0.009 in the alpine and temperate biotas, respectively, p < 0.01) in the same biota.

In the structural equation model (SEM), plant community dissimilarity (1-similarity) rather than its similarity was used to parallel to geographic distance and environmental distance. As is shown in *Figure 4* and *Figure 4—figure supplement 1*, SEM results showed that long-term environmental variables, including soil properties (SOC, TN, TP, and pH) and climate factors (MAT and MAP), had a strong influence on the plant community; while the short-term environmental variables (i.e., SWC, DOC, DON, $NH_4^+$, $NO_3^-$, and AP) hadn't. In the pairwise sites between alpine cross temperate biotas, SEM (*Figure 4*) showed that prokaryotic community similarity was mainly affected by geographic distance (r = 0.388 and 0.320 in top- and subsoils, respectively), relatively long-term environmental distance (r = –0.171 and –0.130 in top- and subsoils, respectively), plant community dissimilarity (r = –0.124 and –0.065 in top- and subsoils, respectively), and relatively short-term environmental distance (r = –0.046 and –0.069 in top- and subsoils, respectively).

Within each biota, geographic distance only had a direct effect on topsoil prokaryotic community similarity (r = −0.275) in the temperate grassland, while its effect in topsoil of the alpine biota was only indirect through relatively short-term (r = 0.140) and long-term environmental distances (r = 0.222), as well as plant community dissimilarity (r = 0.232). In the alpine biota, increases in plant community dissimilarity, relatively long-term and short-term environmental distances, directly decreased the

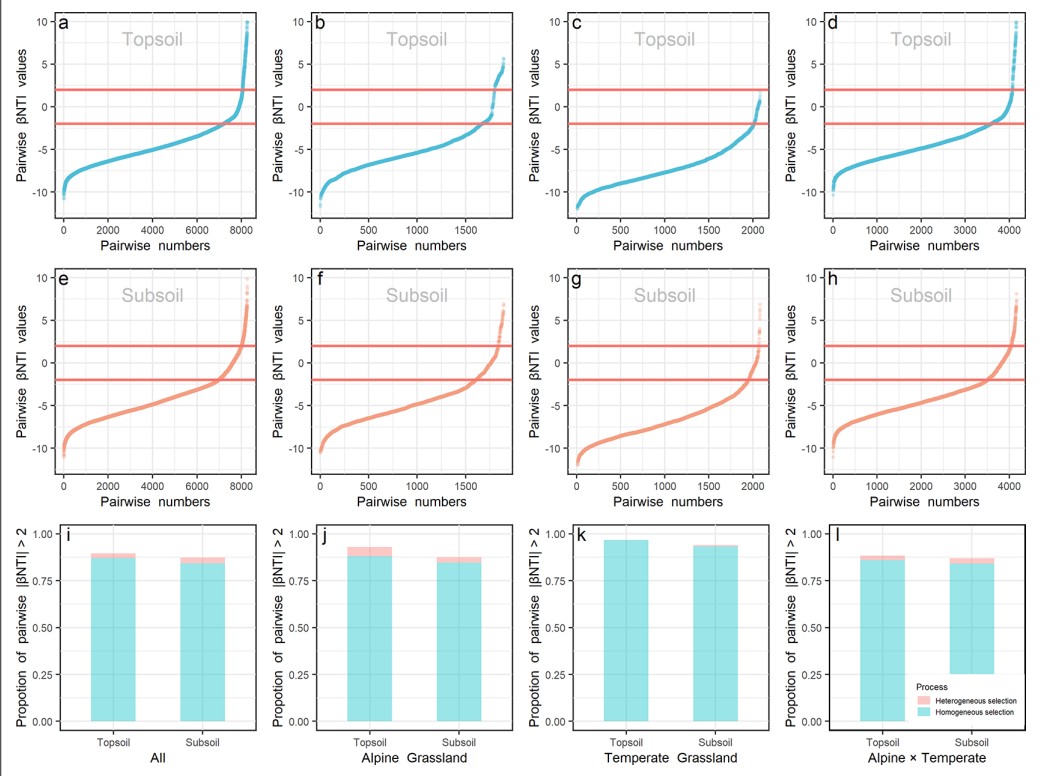

**Figure 3.** βNTI values. βNTI values of prokaryote in the top- (a, b, c, and d) and subsoil (e, f, g, and h) from all sites (a and e), within alpine (b and f) or temperate biota (c and g), and pairwise sites in the alpine cross temperate biotas (d and h) were presented. The proportion of |βNTI| > 2 (deterministic processes) in the top- and subsoils from all sites (i), within alpine biota (j), within temperate biota (k) and pairwise sites in the alpine cross temperate biotas (l) were presented. Light blue and orange colors stand for βNTI values and their proportions in the top- and subsoils, respectively.

The online version of this article includes the following figure supplement(s) for figure 3:

**Figure supplement 1.** Null model analysis for community similarity.

**Figure supplement 2.** The immigration rate (m) of soil prokaryote community in topsoil and subsoil of the alpine and temperate grassland biomes.

---

similarity of soil prokaryotic community. The explained variances of soil prokaryotic community similarity were 26.8% and 21.3% in the top- and subsoils, respectively (*Figure 4—figure supplement 1*). In the temperate biota, other than the direct effect of geographic distance, plant community dissimilarity, relatively long-term and short-term environmental distances, also affected topsoil prokaryotic community similarity directly. In the subsoil of the temperate biota, prokaryotic community similarity was not significantly correlated with any factors and the explained variance was merely 0.2%.

## Discussion

'Everything is related to everything else, but near things are more related to each other' is termed as 'the first law of geography' (*Tobler et al., 1970*). Ecologists and biogeographers refer it to the negative relationship between community similarity and distance as a geographic distance-decay relationship (*Nekola and White, 1999*; *Wang et al., 2016*). Though being regarded as a principal generalization, the geographic distance-decay relationship was denied and overturned in this study as prokaryotic community similarity increased over geographic distance after tipping points of 1920–1760 km when across biotas. This finding is contradictory with most previous studies, including a report that was conducted even at the similar scale of 4000 km to ours but within a single biota (temperate biota) (*Wang et al., 2017a*). Consistently, when within a single biota of alpine or temperate grassland, the distance-decay relationship was still valid in this study as well.

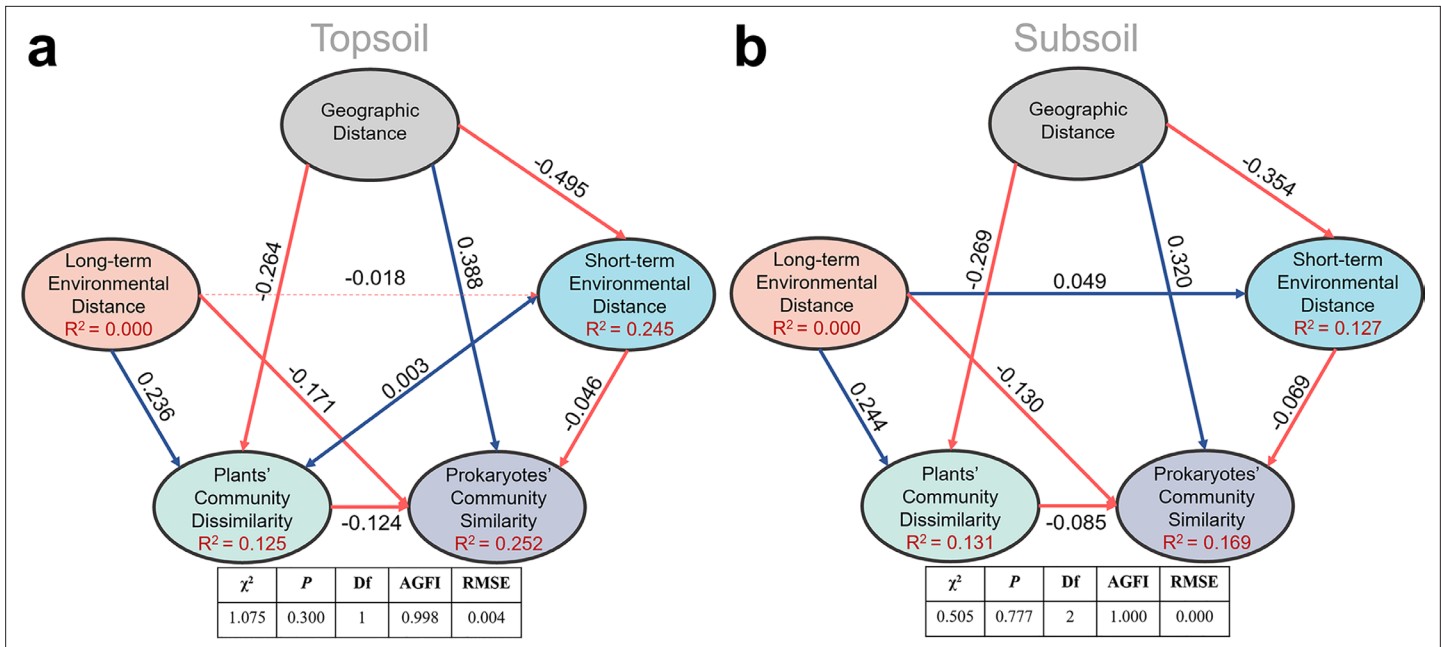

**Figure 4.** Structure equation model to quantify effects of geographic distance, relatively long-term and short-term environmental distance, and plant community dissimilarity on soil prokaryotic community similarity in the pairwise sites between alpine cross temperate biotas, either in top- (**a**) or subsoil (**b**). Red and blue lines stand for negative and positive correlations, respectively; bold lines stand for significance at p = 0.05 level.

The online version of this article includes the following figure supplement(s) for figure 4:

**Figure supplement 1.** Structure equation model for geographic distance, long-term environmental distance, short-term environmental distance, and plant community dissimilarity based on Bray-Curtis distance in affecting soil prokaryote community similarity.

The observed U pattern of prokaryotic community similarity increased over geographic distance in this study was likely due to decreased disparities in environmental heterogeneity over geographic distance when across biotas, supported by three lines of evidences. Firstly, over environmental distance, prokaryotic community similarity showed significant decay relationships (***Figure 2—figure supplement 1***) in all sites on a scale up to 4000 km in this study, consistent with previous reports (***Leff et al., 2018***; ***Xing et al., 2019***). 'Everything is everywhere, but environment selects' (EIE) predicts a distance-decay relationship because environmental selection would lead to a larger community dissimilarity with larger geographic distance, which usually implies higher environmental heterogeneity. However, the EIE hypothesis denies dispersal limitation, whose importance has been supported by some evidences of endemic or non-cosmopolitan distribution of certain microorganisms in certain habitats, for example, the cyanobacterial genus *Synechococcus* in mats of hot springs (***Castenholz, 2017***; ***Papke et al., 2003***). Nevertheless, in this study, we found that the similarity of soil prokaryotic community decreased with environmental distance rather than geographic distance when across biotas or after tipping points of 1760–1920 km. Together with other studies discovering the cosmopolitan distribution (***Cho and Tiedje, 2000***; ***DeLong and Pace, 2001***) of prokaryotes especially at higher taxonomic levels, the observation of this study strongly supports the EIE hypothesis as indicating that all or most prokaryotic groups surveyed in this study reached all or most of their potentially suitable habitats.

Second, null model and βNTI analyses demonstrated that the prokaryotic community assembly was dominantly determined by deterministic processes, especially homogeneous selection referring that similar habitats (environment) harbor similar prokaryotic communities (***Su et al., 2020***). The role of environment filtering, including biotic and abiotic factors, on microbial community assembly has been widely reported at the scale of hundreds to thousands of kilometers (***Shi et al., 2018***; ***Feng et al., 2019***; ***Chu et al., 2016***). This line of evidence also confirms the importance of environmental heterogeneity in determining the observed geographic pattern of soil prokaryotic community. In this study, prokaryotic composition dissimilarity in top or subsoil across the alpine and temperate biotas was not correlated with geographic variables by partial Mantel test, but with certain environmental

variables. These phenomena indicate that influences of these environmental variables overwhelmed the geographic 'distance' effect on shaping the microbial community composition when across biotas. Consistently, even clear biographic patterns of soil microbial communities over space (e.g. taxa-area relationship) reported in previous studies were also mainly attributed to environmental heterogeneity (*Horner-Devine et al., 2004*; *Ranjard et al., 2013*) rather than geographic distance (*Horner-Devine et al., 2004*).

Third, the similarity of relatively short-term environmental variables exhibited a U shape pattern over geographic distance in all sites on a scale up to 4000 km, while that of relatively long-term environmental variables did not, indicating that relatively short-term environmental variables may be responsible for the homogeneous selection shaping the U shape pattern of prokaryotic community over geographic distance. Environmental variables measured in this study were separated into relatively long-term and short-term environmental variables, judged by the dynamic time (*Zhou et al., 2021*). Consistently, SEM analyses also revealed a significant direct effect of relatively short-term environmental distance on prokaryotic community similarity when across biotas. The partial Mantel test further demonstrated that SWC and dissolved nutrients (DOC and DON) were the primary short-term environmental variables in the significant relationship between prokaryotic community similarity and geographic distance. The effects of water on the microbial community have been widely reported (*Suseela et al., 2012*; *Moyano et al., 2013*), as SWC could determine soil texture, bulk density, oxygen availability, and connectivity within soils (*Kaye and Hart, 1997*; *Brockett et al., 2012*; *Månsson et al., 2014*), which can vitally influence soil microbial community composition (*Li et al., 2017*) and microbial basal respiration in the semi-arid area (*Wichern and Joergensen, 2009*). SWC also influences microbial communities through changing nutrient availability. Plenty of studies found that DOC and DON affected the distribution pattern of soil microbes (*Chu et al., 2016*; *Hu et al., 2014*; *Chen et al., 2014*; *Zeng et al., 2016*; *He et al., 2016*). Compared to other nutrients, DOC and DON can be utilized by microbes more directly and easily to provide energy and nutrients for supporting their growth (*Aldén et al., 2001*; *Drenovsky et al., 2004*).

The sampling transects we used in this study had also been adopted, though within a single biota, region or climate type, in previous studies (*Wang et al., 2017a*; *Chu et al., 2016*). Precipitation (MAP) in our sampling sites tends to decrease from northeast to southwest within the temperate biota, while it tends to decrease from southeast to northwest within the alpine biota. Temperature (MAT) in our sampling sites tends to decrease from southwest to northeast within the temperate biota, while it tends to decrease from southeast to northwest within the alpine biota. However, in the topsoil within the temperate biota or both top- and subsoils within the alpine biota, MAP rather than MAT was responsible for the significant decay relationship between prokaryotic community similarity and long-term environmental distance. These phenomena indicated that the spatial pattern of soil prokaryotic community was controlled by long-term precipitation condition rather than temperature within the alpine or temperate biota. Interestingly, though there was no significant correlation between prokaryotic community similarity and MAT or MAP when across biotas, the importance of SWC and dissolved nutrients were discovered.

In addition to abiotic factors, plant community attributes (*de Vries et al., 2012*; *Grayston et al., 1998*; *Singh et al., 2004*; *Berg and Smalla, 2009*; *Hiiesalu et al., 2014*), especially plant species identity (*Leff et al., 2018*; *Peay et al., 2013*; *Prober et al., 2015*), may also be important for the U-shape pattern of prokaryotic community similarity. Our results demonstrated that soil prokaryotic community similarity decreased over the plant community dissimilarity in all sites on a scale up to 4000 km. Moreover, SEM also revealed the significantly direct effect of plant community dissimilarity on prokaryotic community similarity in pairwise sites between alpine cross temperate biotas. Plant community composition and diversity can affect soil prokaryotic communities through altering the quality and quantity of organic matter input to soils by the forms of litterfall and root exudates (*van der Heijden et al., 2008*). Plants exude a substantial proposition (11–40%) of photosynthesis-derived carbon (*Badri and Vivanco, 2009*), including sugars, amino acids, organic acids, fatty acids, and secondary metabolites (*Badri and Vivanco, 2009*; *Bais et al., 2006*; *Baetz and Martinoia, 2014*). Their compounds in exudation can attract beneficial microorganisms deliberately and influence the assembly of soil microbiomes to promote plants' adaptation to the surrounding environment (*Bulgarelli et al., 2013*; *Hooper et al., 2000*; *Wardle, 2006*; *Mellado-Vázquez et al., 2016*; *Jones, 1998*; *Zhalnina et al., 2018*).

In the topsoil, the turnover rate of prokaryote was higher in temperate than alpine biota (*Figure 1*). We found that the immigration rate (m) of topsoil prokaryotes was higher in the temperate biota than that of the alpine biota (*Figure 3—figure supplement 2*), indicating a weakened dispersal limitation that may be responsible for the higher similarity of prokaryotic community in the temperate biota (*Hanson et al., 2012*; *Zhou and Ning, 2017*). The effects of dispersal limitation on microbial communities (*Whitaker et al., 2003*; *Horner-Devine et al., 2004*; *Martiny et al., 2011*) were dependent on ecosystems or environmental habitats (*Chase, 2007*; *Evans et al., 2017*; *Louca et al., 2016*). Harsh environments (*Zhou and Ning, 2017*; *Zhang et al., 2019*; *Young et al., 2013*) with low temperature, high UV, and complex mountain terrain in the alpine biota on the Tibet Plateau would not be conducive for soil prokaryote to disperse. In contrast, the temperate biota has benign temperature, low UV, and better landscape connectivity to promote the spatial dispersal of microorganisms.

Notably, within the temperate biota across 1661 km, prokaryotic community similarity did not change over geographic distance in the subsoil. Moreover, subsoil prokaryotic community similarity was not linked with plant community dissimilarity (*Figure 1—figure supplement 3*) and long-term environmental distance (*Figure 2—figure supplement 1*). It was correlated weakly with the short-term environmental distance based on the correlation test ($R^2 = 0.011$) and SEM ($r = -0.002$). These phenomena denied the possibility that prokaryote community assembly was dependent on plant community or environment heterogeneity in subsoil of the temperate biota. Cases of no distance-decay relationship for microorganisms have also been reported previously (*Zhou et al., 2008*; *Cox et al., 2016*; *Davison et al., 2015*), possibly explained by factors like the paleogeographic history (*Cox et al., 2016*; *Davison et al., 2015*) and plant-dependent (*Davison et al., 2015*).

## Conclusion

This study provides a systematical analysis of the spatial pattern of soil prokaryotic communities in the northern grassland of China. Soil prokaryotic similarity exhibited a U-shape distribution pattern over geographic distance at a scale of up to 4000 km. This finding overturns the well-accepted geographic distance-decay relationship, which was still valid in the top- or subsoils within the alpine biota and only in the topsoil within the temperate biota. Despite different climate and ecosystem types in the alpine and temperate biotas, habitats far more apart when across biotas were more similarly as revealed by the U-shape pattern for short-term environmental variables over geographic distance. Consistently, deterministic processes were found to dominate the soil prokaryotic community assembly by null model and βNTI analyses, and further partial Mantel analysis revealed that SWC and dissolved nutrients (DOC and DON) may be responsible for the decay relationship of prokaryotic community similarity over environmental distance, overturning the geographic distance-decay relationship.

## Materials and methods
### Study sites and field sampling

The alpine grassland in the region of Qinghai-Tibet Plateau belongs to the Qinghai Tibet Plateau biota, and the temperate grassland in the region of Inner Mongolia belongs to Eurasian grassland biota, according to the Chinese floristic geography (*Wu, 2019*). A total of 129 sites and 258 samples were collected from the northern grassland of China. Among them, 128 samples from 64 sites (red dots in *Figure 1—figure supplement 1*) were collected from July 29 to August 14, 2014, in the Qinghai-Tibet Plateau alpine biota, China. Alpine biota sample sites covered a variety of alpine ecosystems, including alpine meadow, alpine steppe, alpine desert, and shrub; 130 samples from 65 sites (blue dots in *Figure 1—figure supplement 1*) were collected from September 10–24, 2015, in the temperate biota, in Inner Mongolia, China. Temperate biota sample sites covered three types of temperate grassland ecosystem, namely temperate meadow, temperate steppe, and temperate desert. The distance between each two adjacent sample sites was no less than 60 km and removed from potential human interference such as towns, villages, and roads.

A GPS (global positioning system) was used to record the geographic coordinates and altitudes of each sample site. Five plots ($1 \times 1$ m$^2$) were selected randomly at each site and the distance between adjacent plots was more than 10 m. In each plot, plant species were identified and their aboveground parts were collected separately by species. After being delivered to the laboratory, the aboveground plant part of each species was dried at 65°C for 72 hr and weighted as the aboveground biomass

(AGB, g/m$^2$). The community AGB in each plot was calculated by the sum of AGBs for all collected species in that plot. After removing plant ABG and litter, three topsoil (0–5 cm in depth) and subsoil (5–20 cm in depth) cores (7 cm in diameter) were randomly sampled within each plot.

Topsoil or subsoil samples from each plot were pooled and then sieved through a 2 mm mesh, and the roots were selected as belowground biomass (BGB). Sieved topsoil or subsoil samples were divided into two subsamples. One part was stored at room temperature and dried in the shade for measuring physical and chemical properties. The other part was stored at approximately 4°C in the field by a mobile refrigerator, delivered with dry ice to the laboratory in Beijing, and finally frozen at –80°C in a freezer before DNA extraction.

## Soil properties

Soil pH was measured by pH meter (STARTER3100, Ohaus Instruments Co., Ltd, Shanghai, China) with a 1:5 of soil water ratio (5 g soil: 25 mL ddH$_2$O). SWC was measured by ovening fresh soil samples at 105°C for 24 hr. SOC was measured by a TOC analyzer (Liqui TOC II; Elementar Analysensysteme GmbH, Hanau, Germany). Soil TN was measured on an auto-analyzer (SEAL Analytical GmbH, Norderstedt, Germany). Soil TP and AP were measured by a UV-VIS spectrophotometer (UV2700, SHIMADZU, Japan). Nitrate-N (NO$_3^-$) and ammonium-N (NH$_4^+$) were extracted with 2 M KCl (soil mass to solution ratio of 1:5) and then analyzed on a continuous-flow ion auto-analyzer (SEAL Analytical GmbH, Norderstedt, Germany). Soil DOC and DON were measured on a TOC Analyser (Liqui TOC II; Elementar Analysensysteme GmbH, Hanau, Germany). Plant AGB and BGB were measured after oven drying at 65°C for 72 hr. MAT (1980–2014) and MAP (1980–2014) of each study site were obtained from 'China Meteorological Data Service Center' (CMDC: https://data.cma.cn/) by latitude and longitude.

## Microbial analysis

Soil genomic DNA was extracted from 0.25 g frozen soil three times at each soil layer at each site and then mixed into one DNA sample using PowerSoil DNA Isolation Kits (MO BIO Laboratories, Carlsbad, CA). The quality of extracted DNA was assessed based on OD 260/280 and 260/230 nm absorbance ratios by NanoDrop (2000) spectrophotometer (NanoDrop Technologies Inc, Wilmington, DE,). Primer pair 515 F (5'-GTGYCAGCMGCCGCGGTA-3') and 909 R (5'-CCCCGYCAATTCMTTTRAGT-3') was selected to amplify the V4-V5 region of 16S rRNA and the target fragment length was 374 bp, and the 12 bp barcode was added at the end of 5' of 515F. A 50 μL PCR system was configured in 0.2 mL tube, including 2 μL template DNA diluent, 4 μL dNTP, 4 μL Mg$_2^+$, 5 μL Buffer, 0.5 L Ex Taq enzyme, 1 μL forward primer, 1 μL reverse primer, 32.5 L ddH$_2$O. The PCR procedure was performed as follows: predenaturation at 95°C for 10 min, 30 PCR cycles (deformation at 94°C for 30 s, annealing at 53°C for 25 s, extension at 68°C for 45 s), and a final extension at 72°C for 10 min. The PCR products were purified by 1% agarose gel using GeneJET Gel Extraction Kit (Thermo Scientific, Lithuania, USA). The purified DNA was tested by NanoDrop (2000) spectrophotometer (NanoDrop Technologies Inc, Wilmington, DE,). All purified DNA samples were mixed in 100 ng before database construction and sequencing, which was performed by Illumina Miseq in Chengdu Biology Institute.

The MiSeq raw data was analyzed by UPARSE pipeline with USEARCH 8 software to obtain an OTUs table. The double ended sequencing was performed by the Illumina Miseq platform with 2 × 250 bp V2 Kits at Chengdu Institute of Biology, Chinese Academy of Sciences. Raw reads generated from the Miseq paired-end sequencing were merged by Fast Length Adjustment of Short reads (FLASH) (*Magoč and Salzberg, 2011*). The fastq sequencing files were combined into one new fastq file. By USEARCH v8.0.1623 (*Edgar, 2013*), barcode sequences were replaced by new labels, and the forward and backward primer sequences were removed by the command of -fastq_filter. The unique representative sequences were selected by the command of -fastx_uniques. After discarding singletons, OTUs were clustered at the 97% threshold and chimeras were removed simultaneously by the command of -cluster_otus. The command of -usearch_global was then used to map the fastq file into an OTU table. Each OTU was annotated by Mothur v1.27 (*Schloss et al., 2009*) with classify. seqs command, and sliva.nr_v128.align was selected as the reference database. The OTU table was resampled to the same sequence (*Figure 1—figure supplement 4*) before further analysis by R 3.5.0 with the resample package.

## Statistical analysis

To compare the soil bacterial samples from different climate regions, we divided the soil samples into alpine samples and temperate samples according to collection sites. The altitude of the alpine biota sampling sites ranged from 2796 to 4891 m, and that of temperate biota was from 10 to 1796 m. According to the sampling position in the soil layers, samples were divided into topsoil (0–5 cm) samples and subsoil (5–20 cm) samples. The geographic distance (km) between each two sites was calculated based on their longitude and latitude using the 'distGeo' function of 'geosphere' package in R program, which considers spheric deviations. Plant communities were classified into four functional groups (grasses, sedges, legumes, and forbs) and plant communities' similarity and dissimilarity were calculated based on Bray-Curtis distance by the vegan package of R. The relatively long-term and short-term environmental variables were defined based on whether they remain stable within 1 year or not. For example, SOC is believed to be stable in natural grasslands for years (*Paul et al., 2001*), thus being divided into the relatively long-term environmental variables; while AP (*Chen et al., 2003*), DOC (*Don and Kalbitz, 2005*), and DON (*Retelletti Brogi et al., 2019*) are reported to be dynamic within 1 year, thus being divided into the relatively short-term environmental variables. This division has been used previously (*Zhou et al., 2021*). In this study, relatively long-term environmental variables included MAP, MAT, pH, SOC, TN, and TP, while relatively short-term environmental variables included SWC, AP, DOC, DON, $NH_4^+$, and $NO_3^-$.

The Bray-Curtis similarity and dissimilarity of the prokaryotic community were calculated using OTU tables resampled to a minimum number of sequences from each sample (7500 in this study). The Mantel test and partial Mantel test based on a Pearson correlation were used to test the relationship of soil prokaryotic similarity, geographic distance, and long-term multiple environmental variables or short-term multiple environmental variables. The turnover rate was estimated by the slope of the linear regression model based on the least square method. The tipping point was calculated by the function of $d(Y)/d(x) = 0$ in binomial function. Pearson correlation was used to test the relationship of soil prokaryotic diversity with environmental variables.

Null model (*Nj and Gr, 1996*) and βNTI analyses (*Fine and Kembel, 2011*) were used to distinguish different ecological processes, including deterministic processes (homogenous selection and heterogeneous selection), random dispersal (homogenous dispersal, dispersal limitation), drift, and diversification (*Zhou and Ning, 2017*). Null model analysis was performed by comparing the Bray-Curtis similarity of observed communities and randomly permutated communities using R3.5.0 with package vegan and picante. The |βNTI| > 2 means community was constructed by deterministic processes, and βNTI < –2 means homogenous selection plays a major role, while βNTI > + 2 means heterogenous is more important. The –2 < βNTI < + 2 means stochastic processes determined community succession (*Dini-Andreote et al., 2015*). A βNTI analysis was performed by R 3.5.0 with the ape package. The estimation of immigration rate (m) was calculated by TeTame 2.0 (*Chave and Jabot, 2008*) based on Hubbell's neutral theory of biodiversity (*Hubbell and Borda-de-Água, 2004*). Parameter estimation was rigorously performed by maximum-likelihood using the sampling formula developed by *Etienne, 2005*; *Etienne and Alonso, 2005*; *Etienne et al., 2006*; *Etienne and Olff, 2005*. This model is seen as a potentially useful null model in ecology; in this model, the species relative abundances in a guild are determined by two parameters, namely θ and m. The θ governs the appearance of a new species in the regional species pool, and m governs immigration into local communities of individuals from the regional species pool. We further used SEM to disentangle the causal pathways through which geographic distance, short-term environmental distance, long-term environmental distance, and plants' community dissimilarity influence soil prokaryotic similarity. SEM in this study is implemented by AMOS software.

## Acknowledgements

This work was financially supported by the Strategic Priority Research and Program A (XDA20050104) of the Chinese Academy of Sciences (CAS), National Natural of China, Science Foundation (42041005), The Second Tibetan Plateau Scientific Expedition and Research (STEP) program (Grant No. 2019QZKK0304), Strategic Priority Research Program A (XDA1907304) and Program B (XDB15010201) of CAS, CAS 'Light of West China' Program, and Sanjiangyuan National Park Joint Program (LHZX-2020-02-01).

# Additional information

## Funding

| Funder | Grant reference number | Author |
|---|---|---|
| Chinese Academy of Sciences | Strategic Priority Research and Program A (XDA20050104) | Kai Xue Yanfen Wang |
| National Natural Science Foundation of China | 42041005 | Yanfen Wang Kai Xue |
| Ministry of Science and Technology of the People's Republic of China | The Second Tibetan Plateau Scientific Expedition and Research program (2019QZKK0304) | Yanfen Wang Kai Xue |
| Chinese Academy of Sciences | Strategic Priority Research Program A (XDA1907304) | Kai Xue |
| Chinese Academy of Sciences | Strategic Priority Research Program B (XDB15010201) | Yanfen Wang |
| Chinese Academy of Sciences | Light of West China | Kai Xue |
| Chinese Academy of Sciences and People's Government of Qinghai Province | Sanjiangyuan National Park Joint Program （LHZX-2020-02-01） | Yanfen Wang |

The funders had no role in study design, data collection and interpretation, or the decision to submit the work for publication.

## Author contributions

Biao Zhang, Data curation, Formal analysis, Investigation, Methodology, Validation, Visualization, Writing - original draft, Writing - review and editing; Kai Xue, Conceptualization, Methodology, Project administration, Validation, Writing - review and editing; Shutong Zhou, Investigation, Validation; Kui Wang, Zongsong Wang, Investigation; Wenjing Liu, Validation; Cong Xu, Qinwei Ran, Visualization; Lizhen Cui, Resources; Linfeng Li, Ronghai Hu, Methodology; Yanbin Hao, Funding acquisition, Investigation; Xiaoyong Cui, Conceptualization, Resources; Yanfen Wang, Conceptualization, Funding acquisition, Methodology, Resources, Supervision

## Author ORCIDs

Biao Zhang ![ORCID] http://orcid.org/0000-0002-3102-8300
Kai Xue ![ORCID] http://orcid.org/0000-0002-5990-4448
Ronghai Hu ![ORCID] http://orcid.org/0000-0001-8041-2483
Yanfen Wang ![ORCID] http://orcid.org/0000-0001-5666-9289

## Decision letter and Author response

Decision letter https://doi.org/10.7554/eLife.70164.sa1
Author response https://doi.org/10.7554/eLife.70164.sa2

# Additional files

## Supplementary files

• Supplementary file 1. Mantel test between the prokaryotic community similarity based on Bray-Curtis index and environment variables.

• Supplementary file 2. Partial Mantel test between prokaryotic community similarity based on Bray-Curtis index and environmental variables.

• Transparent reporting form

## Data availability

Sequencing data has been deposited in the NCBI Sequence Read Archive under accession number: PRJNA729210. Other datasets including OTU table, plant community data and environmental variables are available at https://github.com/zhangbiao1989/Support-files.git copy archived at swh:1:rev:254b3d1fb9b81fae868a594f4fc2c9c3583021d2.

The following dataset was generated:

| Author(s) | Year | Dataset title | Dataset URL | Database and Identifier |
|---|---|---|---|---|
| Xue K | 2021 | Environmental selection overturns the decay relationship of soil prokaryotic community over geographic distance across grassland biotas | https://www.ncbi.nlm.nih.gov/bioproject/PRJNA729210 | NCBI BioProject, PRJNA729210 |

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
