## [Editor Report]

With a massive and extensive survey of soils and prokaryotic communities, this study by Zhang et al. explores the distance-decay relationship in grassland biotas. Their results show that in contrast to the traditional linear relationship within the same biome, a significant U-shape pattern was found across two grassland types. Even more intriguingly, further analysis showed that a homogeneous selection process instead of geographic distance effects may contribute to the soil prokaryotic community patterns. This study will inspire biogeographers, and ecosystem and community ecologists interested in macroecological rules.

---

## [Decision Letter]

**Decision letter after peer review:**

Thank you for submitting your article "Homogeneous environmental selection overturns distance-decay relationship of soil prokaryotic community" for consideration by *eLife*. Your article has been reviewed by 2 peer reviewers, and the evaluation has been overseen by a Reviewing Editor and Detlef Weigel as the Senior Editor. The following individual involved in review of your submission has agreed to reveal their identity: Ye Deng (Reviewer #2).

Essential revisions:

If you accept to undertake a revision, please read carefully the extensive and detailed comments made by the reviewers. You will need to provide a strong response letter assessing whether you agree with the comments (and steps made to improve your manuscript) or disagree with any of them.

*Reviewer #1 (Recommendations for the authors):*

1. The geographic data should be included in the Mantel test, e.g. geographic distance and altitudes. Please add relevant analysis to the contribution of these geographical variables.

2. Descriptions for the sampling efforts and bioinformatics processing. Please add more details. How to investigate the plant community and measure the plant aboveground biomass? How to treat the raw sequencing data with the pipeline and the deal with the singletons? Basically, rarefaction curve should be added to supporting the enough sequencing depth.

3. Data availability. The authors stated the raw sequencing data was archived in NCBI with the bioproject ID PRJNA729210. However, the data was not released yet and it is hard to know the data accessibility and the data format. Moreover, the data affiliated to the analysis should be public as well, e.g. the OTU table, plant community data and environmental variables, to support the robustness of the repeated analysis.

*Reviewer #2 (Recommendations for the authors):*

I have several serious concerns about data analyses and interpretation, which should be addressed before the manuscript is suitable for publication.

1. Lines 24-25: The opening statement of the Abstract is dubious. Both biomes belong to grassland biomes, with one located in alpine regions and the other one located in temperate regions. With the same rationale, many publications have fulfilled such a category if samples of the same kind (e.g., forest, wetland, and desert) are taken across different altitudes, latitudes, and longitudes. Using references in this manuscript as examples, references #1, 17, and 30 could be regarded as studies of different biomes, which argue against the opening statement.

2. Lines 29-31: It is a major conclusion of the manuscript, coined in the title. Thus, it is important to use multiple methods to cross-validate the conclusion. A major limitation of this study is the reliance on a single, new data analysis method that has not been widely evaluated.

3. Lines 54-56: It is absolutely incorrect. "everything is everywhere, but environment selects" predicts a significant distance-decay relationship because environmental selection leads to larger community dissimilarity with larger distance. Even if this study did not observe a significant distance-decay relationship, it still observes a relationship between community dissimilarity and environmental distance, which supports "everything is everywhere, but environment selects".

4. Line 90: Since the Materials and methods section is at the end of the manuscript, it will help if the authors can provide depth information here or in the Results.

5. Lines 148-151: It is difficult to believe this result. Common sense is that mean annual precipitation and mean annual temperature should change substantially over geographical distance. If the result is reliable, it suggests that the authors have taken samples in a very unusual transect, casting doubt on whether their observations can be generalized.

6. Line 204: Similar to my comment on Line 90. The definition of homogeneous selection should be introduced here or earlier. Not many readers know about homogeneous selection.

7. Line 269: References 50 and 51 are incorrectly cited here.

8. Lines 332-333: There is a grammar issue here and elsewhere.

9. Line 340: I am not sure whether the study is limited in the northern grasslands of China because Figure S1 clearly shows that at least one-third of the 258 samples are collected in southwestern China.

---

## [Author Response]

Reviewer #1 (Recommendations for the authors):1. The geographic data should be included in the Mantel test, e.g. geographic distance and altitudes. Please add relevant analysis to the contribution of these geographical variables.

Thanks for pointing this issue out! We have performed Mantel and partial Mantel tests between the prokaryotic community composition and geographic variables, e.g. geographic distance and altitude. By Mantel test (please see Supplementary file 1), the prokaryotic composition dissimilarity in top or subsoil was significantly correlated with geographic distance or altitude within the alpine or temperate biota and across the alpine and temperate biotas, except that the subsoil prokaryotic composition dissimilarity within the temperate biota was not linked to geographic distance. However, by partial Mantel (please see Supplementary file 2), the prokaryotic composition dissimilarity in top or subsoil was not correlated with geographic variables when across biotas. These phenomena indicate that influences of these environmental variables overwhelmed the geographic “distance” effect on shaping the microbial community composition when across biotas. Consistently, even clear biographic patterns of soil microbial communities over space (e.g. Taxa-area relationship) reported in previous studies were also mainly attributed to environmental heterogeneity (Horner-Devine et al., Nature, 2004;432(7018):750-3, doi: 10.1038/nature03073; Ranjard et al., Nat Commun, 2013;4, doi: 10.1038/ncomms2431) rather than geographic distance (Horner-Devine et al., Nature, 2004;432(7018):750-3, doi: 10.1038/nature03073). The corresponding contents have been added in Table 1, Supplementary file 2, Line 177-182 for Results and Line 305-310 for Discussion of new manuscript.

2. Descriptions for the sampling efforts and bioinformatics processing. Please add more details. How to investigate the plant community and measure the plant aboveground biomass? How to treat the raw sequencing data with the pipeline and the deal with the singletons? Basically, rarefaction curve should be added to supporting the enough sequencing depth.

Thanks for pointing these issues out! We have added detailed descriptions for the field sampling efforts and bioinformatics processing.

As stated in Line 409-414 of the new manuscript for the plant community investigation and aboveground biomass measurement, “five plots (1×1m^2^) were selected randomly at each site and the distance between adjacent plots was more than 10 m. In each plot, plant species were identified and their aboveground parts were collected separately by species. After being delivered to the laboratory, the aboveground plant part of each species was dried at 65 ℃ for 72 hours and weighted as the aboveground biomass (AGB, g/ m^2^). The community AGB in each plot was calculated by the sum of AGBs for all collected species in that plot”.

As described in Line 459-468 of the new manuscript for the bioinformatics processes, “the double ended sequencing was performed by the Illumina Miseq platform with 2 × 250 bp V2 Kits at Chengdu Institute of Biology, Chinese Academy of Sciences. Raw reads generated from the Miseq paired-end sequencing were merged by Fast Length Adjustment of Short reads (FLASH) (Magoc and Salzberg 2011, Bioinformatics. 2011;27(21):2957-63, doi:10.1093/bioinformatics/btr507). The 258 fastq sequencing files were combined into one new fastq file. By USEARCH v8.0.1623 (Edgar 2013, Nature Methods. 2013;10(10):996, doi:10.1038/nmeth.2604), barcode sequences were replaced by new labels, and the forward and backward primer sequences were removed by the command of -fastq_filter. The unique representative sequences were selected by the command of -fastx_uniques. After discarding singletons, OTUs were clustered at the 97% threshold and chimeras were removed simultaneously by the command of -cluster_otus. The command of -usearch_global was then used to map the fastq file into an OTU table”.

The rarefaction curve (Figure 1—figure supplement 4) has been added. Based on the rarefaction curve, the sequencing depths were enough for most samples, except a few samples with exceptionally high diversity that might not be satisfied by the current sequencing effort, e.g., NM84 and NM80 from the temperate grassland. However, such heterogeneity is part of the natural characteristics of sampled microbial communities in large-scale studies. We would not except it to change our conclusions in this case due to its limited number.

3. Data availability. The authors stated the raw sequencing data was archived in NCBI with the bioproject ID PRJNA729210. However, the data was not released yet and it is hard to know the data accessibility and the data format. Moreover, the data affiliated to the analysis should be public as well, e.g. the OTU table, plant community data and environmental variables, to support the robustness of the repeated analysis.

Thanks for pointing these issues out! We have already released the sequencing data to NCBI with the bioproject ID PRJNA 729210. The OTU table, plant community data and environmental variables are available to the public at https://github.com/zhangbiao1989/Support-files.git.

Reviewer #2 (Recommendations for the authors):I have several serious concerns about data analyses and interpretation, which should be addressed before the manuscript is suitable for publication.1. Lines 24-25: The opening statement of the Abstract is dubious. Both biomes belong to grassland biomes, with one located in alpine regions and the other one located in temperate regions. With the same rationale, many publications have fulfilled such a category if samples of the same kind (e.g., forest, wetland, and desert) are taken across different altitudes, latitudes, and longitudes. Using references in this manuscript as examples, references #1, 17, and 30 could be regarded as studies of different biomes, which argue against the opening statement.

Thanks for pointing this issue out! We have replaced “biome” by “biota” according to the Chinese floristic geography (Wu et al., 2014, Science Press, Beijing. (in Chinese)) in the new manuscript and explained it in Line 395-397 that “the alpine grassland in the region of Qinghai-Tibet Plateau belongs to the Qinghai Tibet Plateau biota, and the temperate grassland in the region of Inner Mongolia belongs to Eurasian grassland biota”.

2. Lines 29-31: It is a major conclusion of the manuscript, coined in the title. Thus, it is important to use multiple methods to cross-validate the conclusion. A major limitation of this study is the reliance on a single, new data analysis method that has not been widely evaluated.

Thanks for pointing this issue out! We have performed the null model analysis and found its results further support our conclusion. As stated in Line 201-208 of the new manuscript, “Null model (Gotelli N.J. and Graves G.R., 2016, Smithsonian Institution Press, Washington, DC) and βNTI (β-nearest taxon index, Fine et al., Ecography, 2011;34(4):552-65, doi: 10.1111/j.1600-0587.2010.06548.x) analyses were used to explore the roles of deterministic and stochastic processes on community assembly. The similarity of observed communities in all samples (Figure 3—figure supplement 1c) or samples in each biota (Figure 3—figure supplement 1a and b) was significantly (P < 0.001) higher than that of permutated communities generated by the null model, revealing the important role of deterministic processes in determining the community assembly. As shown in Figure 3, the range of |βNTI| > 2 also confirmed the importance of deterministic processes in prokaryotic community assembly, and identified the dominant role (>84%) of homogeneous environmental selection”.

3. Lines 54-56: It is absolutely incorrect. "everything is everywhere, but environment selects" predicts a significant distance-decay relationship because environmental selection leads to larger community dissimilarity with larger distance. Even if this study did not observe a significant distance-decay relationship, it still observes a relationship between community dissimilarity and environmental distance, which supports "everything is everywhere, but environment selects".

After careful consideration, we totally agree with the point raised by reviewer 3.

We have deleted the content in Line 54-56 of the last version manuscript, but added corresponding discussion in Line 284-295 and Line 307-310 of the new manuscript: "Everything is everywhere, but environment selects" (EIE) predicts a distance-decay relationship because environmental selection would lead to a larger community dissimilarity with larger geographic distance, which usually implies higher environmental heterogeneity. However, the EIE hypothesis denies dispersal limitation, whose importance has been supported by some evidences of endemic or non-cosmopolitan distribution of certain microorganisms in certain habitats, e.g. the cyanobacterial genus Synechococcus in mats of hot springs (Castenholz, 1978, Mitt Int Ver Limnol 21: 296–315; Papke et al., Environ Microbiol. 2003;5(8):650-9, doi: 10.1046/j.1462-2920.2003.00460.x). Nevertheless, in this study, we found that the similarity of soil prokaryotic community decreased with environmental distance rather than geographic distance when across biotas or after tipping points of 1,760 – 1,920 km. Together with other studies discovering the cosmopolitan distribution (Cho et al., Appl Environ Microb, 2000;66(12):5448-56, doi: 10.1128/aem.66.12.5448-5456.2000; DeLong et al., Systematic Biology, 2001;50(4):470-8, doi: 10.1080/106351501750435040) of prokaryotes especially at higher taxonomic levels, the observation of this study strongly supports the EIE hypothesis as indicating that all or most prokaryotic groups surveyed in this study reached all or most of their potentially suitable habitats”, “Consistently, even clear biographic patterns of soil microbial communities over space (e.g. Taxa-area relationship) reported in previous studies were also mainly attributed to environmental heterogeneity (Horner-Devine et al., Nature, 2004;432(7018):750-3, doi:10.1038/nature03073; Ranjard et al., Nat Commun, 2013;4, doi:10.1038/ncomms2431) rather than geographic distance (Horner-Devine et al., Nature, 2004;432(7018):750-3, doi:10.1038/nature03073)”.

4. Line 90: Since the Materials and methods section is at the end of the manuscript, it will help if the authors can provide depth information here or in the Results.

Thanks for pointing this out! The soil depths of top- and subsoils have been added in Line 90-92 of the new manuscript as “A total of 258 samples were collected from the top- (0-5 cm in depth) and subsoils (5-20 cm in depth) in both the alpine biota on the Qinghai-Tibet Plateau and the temperate biota on the Inner Mongolia Plateau, on a scale up to 4,000 km”.

5. Lines 148-151: It is difficult to believe this result. Common sense is that mean annual precipitation and mean annual temperature should change substantially over geographical distance. If the result is reliable, it suggests that the authors have taken samples in a very unusual transect, casting doubt on whether their observations can be generalized.

Thanks for pointing this out! The sampling transects we used in this study had also been adopted, though within a single biota, region or climate type, in previous studies (Wang et al., 2017, Isme Journal, 11, 1345-1358, doi: 10.1038/ismej.2017.11; Chu et al., Environmental Microbiology, 18, 1523-1533, doi: 10.1111/1462-2920.13236). Precipitation (MAP) in our sampling sites tends to decrease from northeast to southwest within the temperate biota, while it tends to decrease from southeast to northwest within the alpine biota. Temperature (MAT) in our sampling sites tends to decrease from southwest to northeast within the temperate biota, while it tends to decrease from southeast to northwest within the alpine biota. The corresponding description has been added in Line 331-336 of the new manuscript.

6. Line 204: Similar to my comment on Line 90. The definition of homogeneous selection should be introduced here or earlier. Not many readers know about homogeneous selection.

Thanks for pointing this out! The definition of homogeneous selection has been introduced in Line 208-210 as “selection under homogeneous abiotic and biotic environmental conditions leading to more similar structures among communities (MacArthur RH and Wilson EO. 1967)”.

7. Line 269: References 50 and 51 are incorrectly cited here.

Thanks for pointing this issue out! We have corrected the reference in Line 295 (Su et al., Sci Total Environ. 2020;719, doi: 10.1016/j.scitotenv.2020.137479) of the new manuscript.

8. Lines 332-333: There is a grammar issue here and elsewhere.

Thanks for pointing this out! We have corrected the grammar issue in Line 372-375 of the new manuscript as “It was correlated weakly with the short-term environmental distance based on the correlation test (R^2^ = 0.011) and SEM (r = – 0.002). These phenomena denied the possibility that prokaryote community assembly was dependent on plant community or environment heterogeneity in subsoil of the temperate biota”. We also made efforts to improve the language of the whole text.

9. Line 340: I am not sure whether the study is limited in the northern grasslands of China because Figure S1 clearly shows that at least one-third of the 258 samples are collected in southwestern China.

Thanks for pointing this out! As shown in Figure 4 of Bai et al., Journal of the Chinese Academy of Sciences, 2020:35(6) 675-689, doi: 10.16418/j.issn.1000-3045.20200515003, the northern grassland of China mainly refers to the alpine grassland on Qinghai-Tibet Plateau and the temperate grassland on Mongolia Plateau. They account to 80% area of the northern grassland of China (Li, Scientia Agricultura Sinica. 1997:30(6)1-9; Bai et al., Journal of the Chinese Academy of Sciences, 2020:35(6) 675-689, doi: 10.16418/j.issn.1000-3045.20200515003), exactly where we collected samples. Thus, we are confident with the representativeness of our samples for the northern grasslands of China.

References

Bai Yongfei, Zhao Yujin, Wang Yang, Zhou Kailing. Assessment of Ecosystem Services and Ecological Regionalization of Grasslands Support Establishment of Ecological Security Barriers in Northern China. Journal of the Chinese Academy of Sciences. 2020:35(6) 675-689. (In Chinese)

Brogi SR, Jung JY, Ha S-Y, Hur J. Seasonal differences in dissolved organic matter properties and sources in an Arctic fjord: Implications for future conditions. Sci Total Environ. 2019;694.

Chen CR, Condron LM, Davis MR, Sherlock RR. Seasonal changes in soil phosphorus and associated microbial properties under adjacent grassland and forest in New Zealand. Forest Ecol Manag. 2003;177(1-3):539-57.

Chu H, Sun H, Tripathi BM, Adams JM, Huang R, Zhang Y, et al. Bacterial community dissimilarity between the surface and subsurface soils equals horizontal differences over several kilometers in the western Tibetan Plateau. Environ Microbiol. 2016;18(5):1523-33.

Don A, Kalbitz K. Amounts and degradability of dissolved organic carbon from foliar litter at different decomposition stages. Soil Biol Biochem. 2005;37(12):2171-9.

Edgar RC. UPARSE: highly accurate OTU sequences from microbial amplicon reads. Nature Methods. 2013;10(10):996-+.

Fine PVA, Kembel SW. Phylogenetic community structure and phylogenetic turnover across space and edaphic gradients in western Amazonian tree communities. Ecography. 2011;34(4):552-65.

Papke RT, Ramsing NB, Bateson MM, Ward DM. Geographical isolation in hot spring cyanobacteria. Environ Microbiol. 2003;5(8):650-9.

Paul EA, Collins HP, Leavitt SW. Dynamics of resistant soil carbon of midwestern agricultural soils measured by naturally occurring C-14 abundance. Geoderma. 2001;104(3-4):239-56.

Li Bo. The Rangeland Degradation in North China and Its Preventive Strategy. Scientia Agricultura Sinica. 1997:30(6)1-9. (In Chinese)

MacArthur RH, Wilson EO. 1967. The theory of island biogeography. Princeton University Press, Princeton, NJ.

N.J. G, G.R. G. Null models in ecology. Smithsonian Institution Press, Washington, DC. 1996.

Ranjard L, Dequiedt S, Prevost-Boure NC, Thioulouse J, Saby NPA, Lelievre M, et al. Turnover of soil bacterial diversity driven by wide-scale environmental heterogeneity. Nat Commun. 2013;4.

Shi Y, Li YT, Xiang XJ, Sun RB, Yang T, He D, et al. Spatial scale affects the relative role of stochasticity versus determinism in soil bacterial communities in wheat fields across the North China Plain. Microbiome. 2018;6.

Su Y-g, Liu J, Zhang B-c, Zhao H-m, Huang G. Habitat-specific environmental factors regulate spatial variability of soil bacterial communities in biocrusts across northern China's drylands. Sci Total Environ. 2020;719.

Wang XB, Lu XT, Yao J, Wang ZW, Deng Y, Cheng WX, et al. Habitat-specific patterns and drivers of bacterial β-diversity in China's drylands. Isme J. 2017;11(6):1345-58.

Wu ZY, Sun H (2014) The floristic geography of China. In: Floristic and Vegetation Geography of China (ed. Chen LZ). Science Press, Beijing. (in Chinese)

Zhou S, Xue K, Zhang B, Tang L, Pang Z, Wang F, et al. Spatial patterns of microbial nitrogen-cycling gene abundances along a precipitation gradient in various temperate grasslands at a regional scale. Geoderma. 2021;404.